# LCGen: Mining in Low-Certainty Generation for View-consistent Text-to-3D

**Zeng Tao[1], Tong Yang[2], Junxiong Lin[1], Xinji Mai[1], Haoran Wang[1],**
**Beining Wang[2], Enyu Zhou[3], Yan Wang[1],\*Wenqiang Zhang[2,4],\***

[1]Shanghai Engineering Research Center of AI & Robotics, Academy for Engineering
& Technology, Fudan University, Shanghai, China
[2]Shanghai Key Lab of Intelligent Information Processing, School of
Computer Science, Fudan University, Shanghai, China
[3]School of Computer Science, Fudan University, Shanghai, China
[4]Engineering Research Center of AI & Robotics, Ministry of Education, Academy for
Engineering & Technology, Fudan University, Shanghai, China
{ztao19,yanwang19,wqzhang}@fudan.edu.cn

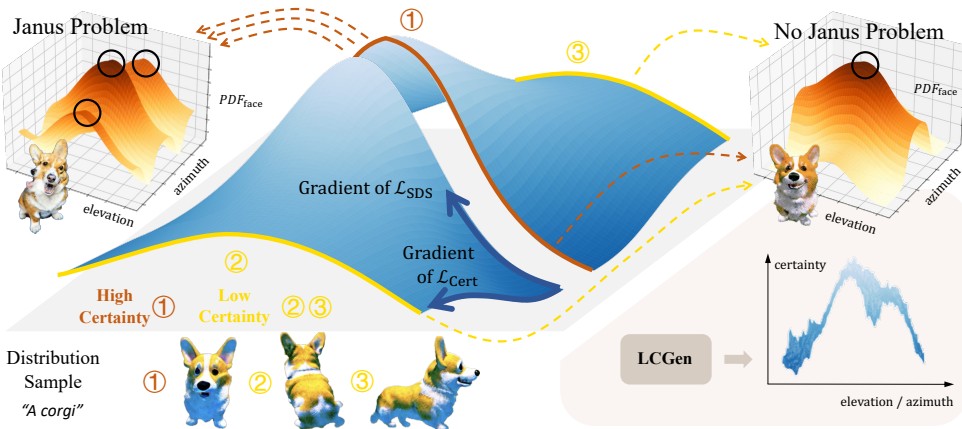

Figure 1: The main cause of the Janus Problem in SDS-based text-to-3D methods is their reliance on high-certainty 2D priors for 3D representation, which can result in heads appearing from multiple viewpoints. To address this, we introduced the LCGen method, using low-certainty generation to align viewpoints with the optimization direction.

## Abstract

The Janus Problem is a common issue in SDS-based text-to-3D methods. Due to view encoding approach and 2D diffusion prior guidance, the 3D representation model tends to learn content with higher certainty from each perspective, leading to view inconsistency. In this work, we first model and analyze the problem, visualizing the specific causes of the Janus Problem, which are associated with discrete view encoding and shared priors in 2D lifting. Based on this, we further propose the LCGen method, which guides text-to-3D to obtain different priors with different certainty from various viewpoints, aiding in view-consistent generation. Experiments have proven that our LCGen method can be directly applied to different SDS-based text-to-3D methods, alleviating the Janus Problem without introducing

---

*Corresponding authors

38th Conference on Neural Information Processing Systems (NeurIPS 2024).

additional information, increasing excessive training burden, or compromising the generation effect. Project page is [here].

# 1  Introduction

At the forefront of the digital domain, considerable advancements have been achieved in converting textual prompts into realistic 2D images, signaling a new epoch of computational creativity [14, 5, 4, 3, 2, 36, 38, 39, 30, 31, 22, 17, 18]. However, transferring such achievements to the 3D domain introduces an additional layer of intricacy. While 3D generation technology [35, 29, 9, 24, 32] is becoming increasingly indispensable across various fields, from virtual reality to architectural design, traditional 3D content generation demands a substantial investment of time and expertise, and the difficulty in acquiring 3D data makes explicit text-to-3D modeling exceedingly challenging. In response, Score Distillation Sampling (SDS) [25], based on 2D lifting, has emerged to simplify and advance the 3D creative process [6, 16, 23, 34]. In text-to-3D tasks, 3D representation (such as NeRF [24]) renders and outputs the corresponding visual image of given camera viewpoints. After that, SDS employs the guidance from priors embedded within pre-trained text-conditioned 2D diffusion models to compute losses for images or latents, iteratively guiding the 3D representation to its optimality.

However, this paradigm carries certain risks, with the Janus Problem [25] standing out as a significant and common issue [33, 37, 40]. This problem occurs when 3D models exhibit multiple, often conflicting viewpoints, resulting in inconsistencies with the original text conditions. As illustrated in Fig. 1, faces appear at various positions within the same 3D object. This is an inevitable result stemming from the inherent nature of SDS-based text-to-3D approaches. Firstly, current SDS-based text-to-3D methods employ discrete viewpoint encoding. Camera perspectives are classified into regions, with each region sharing a uniform view and text guidance. Consequently, the images within each region share the same prior distribution, biasing the 3D representation towards locally optimal synthesis with the highest certainty, as shown in Fig. 2 and Sec. 3. From a global perspective, there is a high probability of bias towards synthesizing heads at multiple biased positions. Secondly, diffusion models lack diverse 3D training data and thus a nuanced understanding of 3D space [19]. For different camera views, there is a tendency to generate images with high certainty that emphasize the most characteristic features of the object [11], such as the head [1]. These intrinsic shortcomings render the Janus Problem a dominant challenge in the text-to-3D process.

Addressing Janus Problem remains a critical challenge, with numerous studies dedicated to mitigating its effects [15, 19–21, 28, 40, 12], like DreamControl [13] and Perp-Neg [1]. However, these methods either require extensive multi-stage fine-tuning or object-specific designs and do not address the fundamental causes of the Janus Problem. In response, we have innovatively modeled and analyzed the underlying causes of the text-to-3D Janus Problem and proposed a novel approach named Low Certainty Generation (LCGen) method.

Specifically, we first analyzed the causes of the Janus Problem through probability modeling. Under the paradigm of discrete viewpoint encoding, viewpoint inputs from the same region possess the same prior distribution. We modeled this distribution and analyzed the relationship between the occurrence of the Janus Problem at various biased positions and the distribution itself. By calculating probabilities, we derived the likelihood of biases occurring at specific positions, as illustrated in Fig. 2 showing probability density peaks at different biases, indicating the Janus Problem.

Thus, addressing the text-to-3D Janus Problem necessitates reevaluating the relationship between views and distributions of guidance, advocating for their decoupling rather than adherence to a shared distribution. Direct explicit modeling of the distribution is quite challenging, so we sought implicit distribution constraints based on certainty learning. Here, we define the probability $C(x_{t-1}|x_t)$ in Eq. 7, estimated at the current timestep, as generation certainty in diffusion. We discovered that different viewpoint images exhibit varying levels of certainty during the diffusion denoising process. For areas rich in object features (such as the front), diffusion model tends to exhibit higher certainty during the denoising process, and conversely lower elsewhere. This is linked to the data bias in pretrained diffusion models [33] (see Appendix. B). Leveraging this characteristic, we constrain the generation of 3D representations so that their input $x_t$ into diffusion model achieves certainty consistent with the viewpoint. Consequently, we obtain a decoupled data distribution for precise distribution localization of separated viewpoints, thus releasing the Janus Problem. Extensive data

analysis and visualization substantiate the scientific validity and effectiveness of LCGen. Our method can be integrated into various SDS-based text-to-3D methods, consistently mitigating the Janus Problem without compromising generative performance.

Our contributions are as follows:

- We model and analyze the Janus Problem in text-to-3D, identifying the fundamental reasons for its occurrence. Our findings indicate that the inevitability of the Janus Problem is associated with the SDS-based text-to-3D framework that employs discrete view encoding and 2D diffusion lifting.
- We specifically develop LCGen, a method that decouples the distribution of viewpoint data from the perspective of generation certainty, thereby guiding precise view localization and effectively mitigating the Janus Problem.
- We conduct extensive data analysis and experiments to demonstrate the scientific validity and effectiveness of LCGen. Our method is transferable and consistently mitigates the Janus Problem across various baselines without compromising the generative quality.

## 2 Background

### 2.1 Diffusion model

Diffusion models [10, 27] have proven a powerful class of generative models, particularly excelling in text-to-image synthesis. Building on the progress made with text-to-image diffusion models, new approaches have been developed to extend these capabilities to 3D content generation.

A diffusion model typically involves a forward process that gradually adds noise to a data sample and a reverse process that aims to reconstruct the original sample by progressively denoising it. Mathematically, the forward process is described as:

$$q(\boldsymbol{x}_t|\boldsymbol{x}_{t-1}) = \mathcal{N}(\boldsymbol{x}_t; \sqrt{1-\beta_t}\boldsymbol{x}_{t-1}, \beta_t\mathbf{I}), \tag{1}$$

where $\mathcal{N}$ is Gaussian distribution, $\beta_t$ are variance terms increasing over time, and $\mathbf{I}$ is the identity matrix. The reverse process, which is more pertinent to generative tasks, is modeled as:

$$p_\Phi(\boldsymbol{x}_{t-1}|\boldsymbol{x}_t) = \mathcal{N}(\boldsymbol{x}_{t-1}; \mu_\Phi(\boldsymbol{x}_t, t), \sigma_t^2\mathbf{I}), \tag{2}$$

where $\mu_\Phi(\boldsymbol{x}_t, t)$ represents the mean learned by the diffusion model $\Phi$, and $\sigma_t^2$ are learned variances.

### 2.2 SDS-based text-to-3D

**Score Distillation Sampling (SDS).** Score Distillation Sampling (SDS) [25] is a novel approach tailored for bridging the gap between 2D image generation and 3D model synthesis. The SDS process utilizes the gradients from a pretrained 2D diffusion model $\Phi$ to guide the generation of 3D model $\Theta$. The key idea is to render 2D projections of a 3D model from various views and adjust the model parameters to maximize the agreement between these projections and the images generated by a text-conditioned diffusion model. Given text prompt $y$, SDS can be described as:

$$\nabla_\Theta \mathcal{L}_{\text{SDS}}(\Theta) = \mathbb{E}_{t,\boldsymbol{\epsilon},c}\left[\Omega(t)(\hat{\boldsymbol{\epsilon}}_\Phi(\boldsymbol{x}_t, t, y^c) - \boldsymbol{\epsilon})\frac{\partial \boldsymbol{x}}{\partial \Theta}\right] \tag{3}$$

where $\hat{\boldsymbol{\epsilon}}_\Phi$ is the noise predicted by $\Phi$, $\boldsymbol{x}$ is the rendered image of view $c$ by $\Theta$, $\Omega$ is weighting factor.

**Variational Score Distillation (VSD).** Building upon SDS, Variational Score Distillation (VSD) [34] introduces a probabilistic framework that treats the problem of text-to-3D synthesis as a distribution optimization task. VSD seeks to create a distribution over possible 3D shapes that is likely under the given text condition, rather than finding a single deterministic shape. VSD can be expressed as:

$$\nabla_\Theta \mathcal{L}_{\text{VSD}}(\Theta) = \mathbb{E}_{t,\boldsymbol{\epsilon},c}\left[\Omega(t)(\hat{\boldsymbol{\epsilon}}_\Phi(\boldsymbol{x}_t, t, y^c) - \hat{\boldsymbol{\epsilon}}_\Theta(\boldsymbol{x}_t, t, c, y))\frac{\partial \boldsymbol{x}}{\partial \Theta}\right] \tag{4}$$

where $\hat{\boldsymbol{\epsilon}}_\Theta(\boldsymbol{x}_t, t, c, y)$ is the noise predicted by rendered images. This approach allows for exploring a richer space of 3D geometries, potentially capturing more complex and diverse features consistent with the textual description.

# 3 Analysis of Janus Problem in Text-to-3D

In this Section, we analyze the reasons for the Janus Problem produced by the SDS-based text-to-3D method. Through modeling, we have identified how the discrete view encoding method leads to shared distributions that cause the Janus Problem. For detailed derivation, please refer to Appendix C. In response to this finding, we have developed the LCGen method, as described in Section 4.

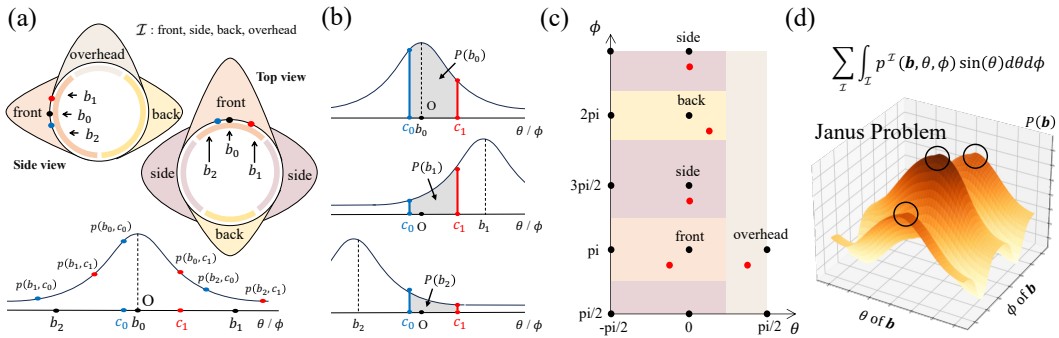

Figure 2: Analysis of the Janus Problem in Text-to-3D. Due to the discrete encoding of viewpoints, there is a high probability of multiple heads appearing at different positions $b$ on the sphere.

Our modeling focuses on two variables: the camera viewpoint parameter $c$ and the position of the head $b$ in the 3D representation. Both variables can be considered as points on a sphere $S^2$, represented by $(\theta, \phi)$. Taking the head position as an example, our modeling yields the probability $P(b)$ of the head appearing at each position $b$ on the sphere. The modeling steps are as follows:

As shown in Fig. 2(a), in the SDS, the sphere $S^2$ is divided into different region intervals $\mathcal{I}$. For each $c$ within $\mathcal{I}$, the same viewpoint text is used, resulting in the same text condition and diffusion guidance (See Appendix. B). We represent the guidance as a superposition of Gaussian distributions with the spherical position as the variable. In one-dimensional Gaussian distribution across $\theta$ or $\phi$ dimension, the probability $p$ of generating a head at position $b$ under the viewpoint $c$ is given by:

$$p^{\mathcal{I}}(b, c) = \sum_i w_i^{\mathcal{I}} \frac{1}{\sqrt{2\pi\sigma_i^{\mathcal{I}2}}} \exp\left(-\frac{(c - b - \mu_i^{\mathcal{I}})^2}{2\sigma_i^{\mathcal{I}2}}\right) \tag{5}$$

where $w_i^{\mathcal{I}}$ are the mixture weights for $i$th Gaussian component in $\mathcal{I}$, $\mu_i^{\mathcal{I}}$ and $\sigma_i^{\mathcal{I}2}$ are the mean and variance of the $i$-th Gaussian component in $\mathcal{I}$, $\sum_i w_i^{\mathcal{I}} = 1$ to ensure that the total probability of the mixture distribution in $\mathcal{I}$ integrates to one over its domain.

As shown in Fig. 2(b), integrating over the viewpoints $[c_0, c_1]$ within $\mathcal{I}$ gives the probability $P$ of generating a head at position $b$ within $\mathcal{I}$ as $P^{\mathcal{I}}(b) = \int_{\mathcal{I}} p^{\mathcal{I}}(b, c)\, dc$.

Now, we extend from one dimension Gaussian distribution to two dimensions $p^{\mathcal{I}}(\boldsymbol{b}, \boldsymbol{c}) = \sum_i w_i^{\mathcal{I}} \frac{1}{\sqrt{(2\pi)^k |\boldsymbol{\Sigma}_i^{\mathcal{I}}|}} \exp\left(-\frac{1}{2}(\boldsymbol{c} - \boldsymbol{b} - \boldsymbol{\mu}_i^{\mathcal{I}})^\top \boldsymbol{\Sigma}_i^{\mathcal{I}-1}(\boldsymbol{c} - \boldsymbol{b} - \boldsymbol{\mu}_i^{\mathcal{I}})\right)$ in Fig. 2(c) and consider spherical integration. the probability of generating a head at position $\boldsymbol{b}$ is given by:

$$P(\boldsymbol{b}) = \sum_{\mathcal{I}} \int_{\mathcal{I}} p^{\mathcal{I}}(\boldsymbol{b}, \boldsymbol{c})\, dA = \sum_{\mathcal{I}} \int_{\mathcal{I}} p^{\mathcal{I}}(\boldsymbol{b}, \theta, \phi) \sin(\theta)\, d\theta\, d\phi \tag{6}$$

where $\mathcal{I}$ represents different regions of the sphere, $dA$ is the differential solid angle element in spherical coordinates, and $\sin(\theta)$ accounts for the area element on the sphere. Each integral $\int_{\mathcal{I}}$ calculates the contribution to $P(\boldsymbol{b})$ from each region $\mathcal{I}$.

Thus, we can use numerical integration techniques to obtain the probability of generating a head at different positions $\boldsymbol{b}$. As shown in Fig. 2(d), probability peaks appear at different positions $\boldsymbol{b}$. This indicates that discrete viewpoint encoding may lead to the generation of heads at different positions in the 3D representation, known as the Janus Problem. Another head is equally likely to appear on the side and back, compromising the realism of the 3D representation. See details in Appendix C.

# 4 LCGen: Low Certainty Generation

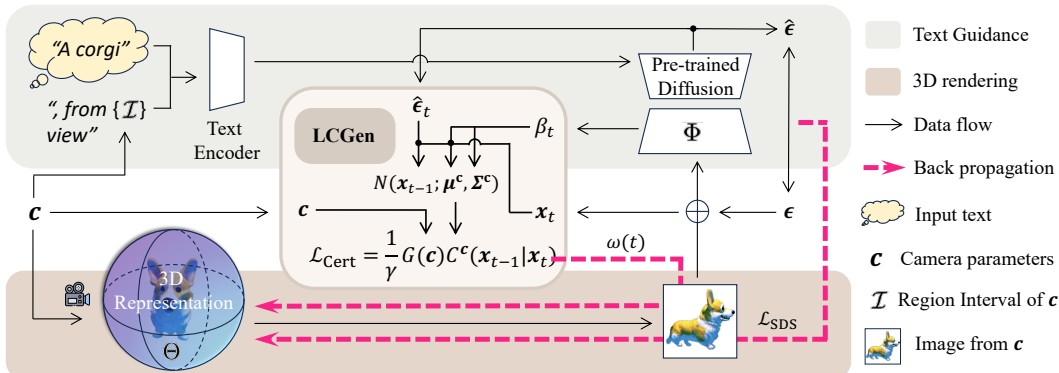

Figure 3: Overview of LCGen. LCGen can be embedded into any SDS-based text-to-3D method, providing different guidance for various viewpoints by constraining the generation certainty, thus alleviating Janus Problem.

From the analysis in Sec. 3, it is evident that the discrete view-dependent text condition can lead the 3D representation to manifest the Janus Problem at different positions $b$ during the synthesis process. Therefore, addressing the relationship between $c$ and guidance presents a viable method for mitigating the Janus Problem. We propose LCGen, which leverages the certainty characteristics to decouple distributions across different viewpoints, thereby mitigating the Janus Problem in the text-to-3D task. Specifically, for $c \in \mathcal{I}$, we constrain the guidance towards $p_t^c(\boldsymbol{x}_t \mid \boldsymbol{c}, y)$ rather than $p_t^{\mathcal{I}}(\boldsymbol{x}_t \mid \boldsymbol{c}, y)$.

**LCGen.** During the diffusion denoising process, different $c$ with different $\boldsymbol{x}_t$ of the same object possess distinct certainty $C(\boldsymbol{x}_{t-1}|\boldsymbol{x}_t)$. We decouple these to guide the synthesis process to generate images that correspond more closely to the desired viewpoint.

In the diffusion model in SDS, each denoising step is considered a probabilistic inference process from the current state $\boldsymbol{x}_t$ to the previous state $\boldsymbol{x}_{t-1}$. This process typically relies on the following assumptions: 1) The noise $\epsilon$ follows a Gaussian distribution, which is estimated by the model at each step. 2) The mapping from $\boldsymbol{x}_t$ to $\boldsymbol{x}_{t-1}$ can be represented using a parameterized Gaussian process.

Assuming we have obtained the prediction of $\hat{\epsilon}_t$ through the diffusion model, we can predict the estimation of the certainty of $\boldsymbol{x}_{t-1}$ given $\boldsymbol{x}_t$. This estimation typically assumes that the certainty follows a Gaussian distribution:

$$C^{\boldsymbol{c}}(\boldsymbol{x}_{t-1}|\boldsymbol{x}_t) \triangleq \mathcal{N}(\boldsymbol{x}_{t-1}; \boldsymbol{\mu}^{\boldsymbol{c}}, \boldsymbol{\sigma}^{\boldsymbol{c}2}) \tag{7}$$

Here, $\boldsymbol{\mu}^{\boldsymbol{c}}$ and $\boldsymbol{\sigma}^{\boldsymbol{c}2}$ represent the mean and variance that guides the U-Net's prediction, thus affecting the synthetic results. $\boldsymbol{\mu}^{\boldsymbol{c}}$ and $\boldsymbol{\sigma}^{\boldsymbol{c}2}$ can be calculated as follows:

$$\boldsymbol{\mu}^{\boldsymbol{c}} = \frac{1}{\sqrt{1-\beta_t}}(\boldsymbol{x}_t - \sqrt{1-\beta_t}\hat{\epsilon}_t), \boldsymbol{\sigma}^{\boldsymbol{c}2} = 1 - \alpha_t \tag{8}$$

Here, $\beta_t$ is the variance parameter at step $t$, $\alpha_t$ is $1 - \beta_t$. The Gaussian distribution parameters guide the Diffusion model's prediction to minimize the Janus Problem at step $t - 1$. The certainty function is defined as:

$$C^{\boldsymbol{c}}(\boldsymbol{x}_{t-1}|\boldsymbol{x}_t) = \frac{1}{\sqrt{2\pi}\boldsymbol{\sigma}^{\boldsymbol{c}}} \exp\left(-\frac{(\boldsymbol{x}_{t-1} - \boldsymbol{\mu}^{\boldsymbol{c}})^2}{2\boldsymbol{\sigma}^{\boldsymbol{c}2}}\right) \tag{9}$$

By constraining $C^{\boldsymbol{c}}(\boldsymbol{x}_{t-1}|\boldsymbol{x}_t)$, we can ensure that different viewpoints have different distributions. We have designed the $\mathcal{L}_{\text{cert}}$ as follows:

$$\mathcal{L}_{\text{cert}} \triangleq \frac{1}{\gamma} \cdot C^{\boldsymbol{c}}(\boldsymbol{x}_{t-1}|\boldsymbol{x}_t) \cdot G(\boldsymbol{c}) \tag{10}$$

where $\gamma$ is normalization constant, $G(\boldsymbol{c})$ is the function of view-based guidance.

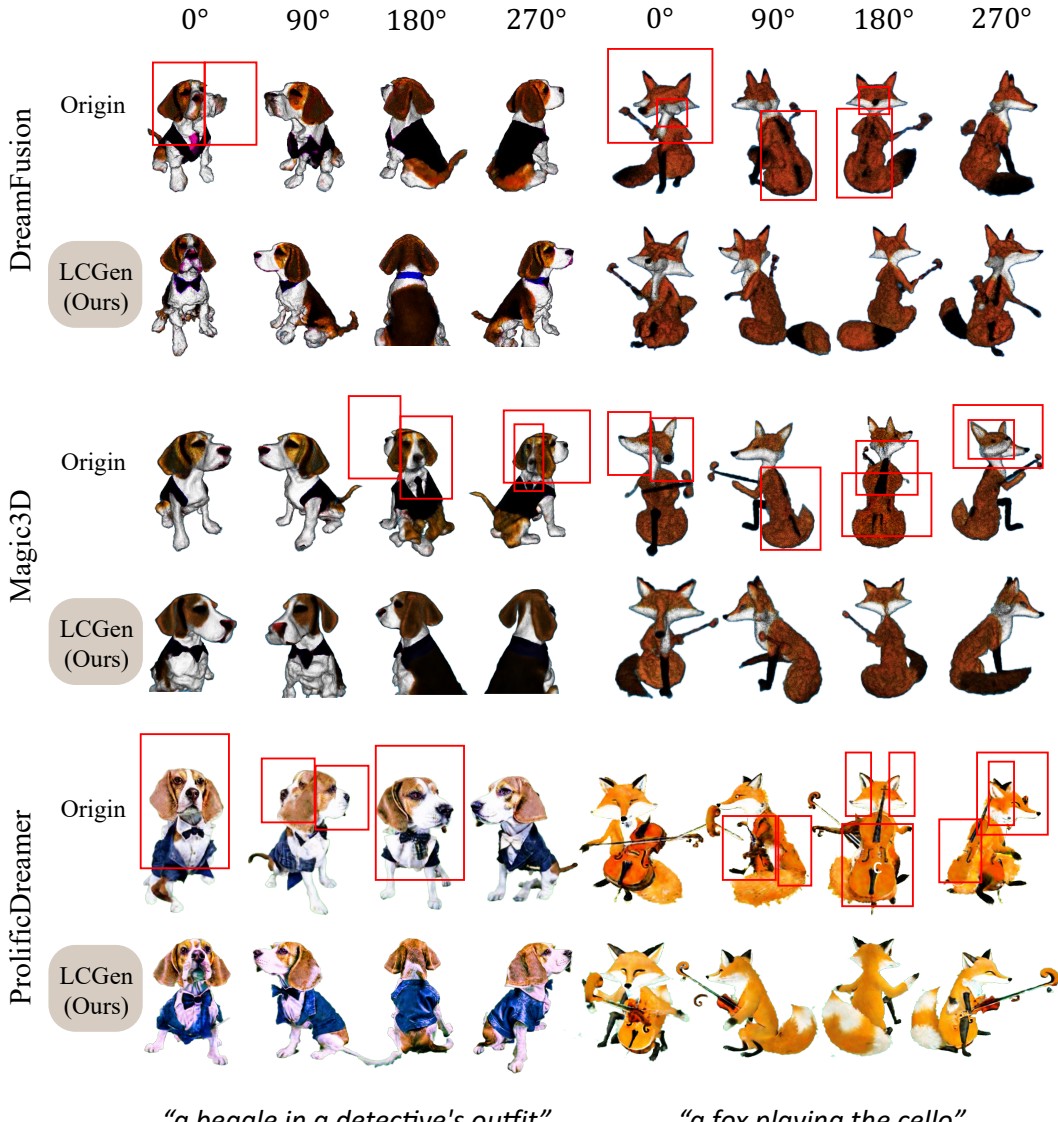

Figure 4: Results of Qualitative Comparison. The areas enclosed in red boxes are where the Janus Problem occurs.

**Back propagation.** Although Eq. 10 specifies the basic form of $\mathcal{L}_{\text{cert}}$, direct backpropagation involves redundancy. In this method, the diffusion model serving as guidance is frozen, and it is unnecessary to calculate gradients for it. We hope to perform backpropagation directly on NeRF. In SDS [25], to simplify the gradient calculation and update processes, researchers simplify the loss function $\mathcal{L}_{\text{SDS}}$, avoiding the complex gradient calculations involved with a frozen diffusion model $\Phi$. The gradients are applied directly to NeRF $\Theta$ and its rendered images $\boldsymbol{x}_t$. In LCGen, we also specifically design the $\mathcal{L}_{\text{cert}}$ to bypass the diffusion model and apply directly in the NeRF flow parameter updates.

For $\mathcal{L}_{\text{cert}}$, we obtain:

$$\frac{\partial \mathcal{L}_{\text{cert}}}{\partial \Theta} = \frac{\partial \mathcal{L}_{\text{cert}}}{\partial \boldsymbol{x}_t} \cdot \frac{\partial \boldsymbol{x}_t}{\partial \Theta} \tag{11}$$

Using the chain rule, we can expand and simplify the first term on the right side in Eq. 11. For details of the process, please refer to the Appendix D. We get:

$$\nabla_\Theta \mathcal{L}_{\text{Cert}}(\Theta) = \mathbb{E}_{t,\boldsymbol{\epsilon},\boldsymbol{c}} \left[ \omega(t) \cdot \frac{1}{\gamma} \cdot C^{\boldsymbol{c}}(\boldsymbol{x}_{t-1}|\boldsymbol{x}_t) \cdot G(\boldsymbol{c}) \cdot \frac{\partial \boldsymbol{x}_t}{\partial \Theta} \right]$$
$$= \mathbb{E}_{t,\boldsymbol{\epsilon},\boldsymbol{c}} \left[ \omega(t) \cdot \mathcal{L}_{\text{cert}} \cdot \frac{\partial \boldsymbol{x}_t}{\partial \Theta} \right] \tag{12}$$

where

$$\omega(t) = -\frac{\boldsymbol{x}_{t-1} - \frac{1}{\sqrt{1-\beta_t}}(\boldsymbol{x}_t - \sqrt{1-\beta_t}\hat{\boldsymbol{\epsilon}}_t)}{\boldsymbol{\sigma}^{\boldsymbol{c}2}} \cdot \frac{1}{\sqrt{1-\beta_t}} \tag{13}$$

By simplifying the calculations, we enable $\mathcal{L}_{\text{cert}}$ to apply directly in the NeRF flow parameter $\Theta$ updates, avoiding the complex gradient calculations of the Unet layer in diffusion model $\Phi$, and maintaining calculation consistency with $\mathcal{L}_{\text{SDS}}$.

## 5 Experiment

In this Section, we apply LCGen to several baseline methods of SDS-based text-to-3D, including DreamFusion [25], Magic3D [16], and ProlificDreamer [34], and conduct corresponding experiments. We also compare with other methods that address the Janus Problem. Sec. 5.2 presents the effects of original methods and LCGen, including qualitative and quantitative assessments. Sec. 5.3 demonstrates the ablation of hyperparameters. Furthermore, Sec. 5.4 presents visualization results of LCGen's impact on generation certainty.

### 5.1 Experiment Settings

We implement original methods and LCGen based on threestudio [8] and a single A100 GPU. In the experiment, we set $G(\boldsymbol{c}) = |\phi|$ and $\gamma$ to 10, and obtained the results after a maximum of 10,000 steps. For the sake of experimental consistency, we have chosen the Stable Diffusion 2.1 base [27] as guidance and NeRF [24] as the 3D representation in the SDS-based method. See Details in Appendix. E and F.

### 5.2 Results of LCGen

**Qualitative Comparison.** We selected two sets of text prompts from the library [25] and conducted experiments on three SDS-based text-to-3D baseline methods, including DreamFusion-sd [25], Magic3D-sd coarse [16], and ProlificDreamer [34], both without and with LCGen, with qualitative results as shown in Fig. 4. It can be observed that, as indicated by the red boxes in Fig. 4, the original methods have a high probability of exhibiting the Janus Problem. For instance, in the first set of examples, the beagle appears with two faces, and in the second set of cases, the fox's front and back both exhibit a cello. As previously analyzed, this is a common issue inherent to SDS-based methods, resulting from the nature of the paradigm. After applying LCGen to each method, the Janus Problem was mitigated, with the generated 3D content exhibiting spatial consistency. In particular, our method has a very strong suppressive effect on the Janus Problem that occurs on the backside of objects.

**Quantitative Comparison.** We have conducted a quantitative analysis of the results from our three sets of baselines, as shown in Fig. 5. The Janus Rate (JR) represents the rate at which the Janus Problem occurs and is used to measure the 3D consistency of the generation method. The CLIP-Score (CS), on the other hand, is a metric for assessing the consistency between the text and the prompt and the generated image. See Appendix. F for details. We selected 30 sets of text prompts from the library and calculate the mean score. It can be observed that our method significantly reduces the probability of the

Figure 5: Results of Quantitative Comparison. ↓ represents that a smaller value is better, while ↑ indicates that a larger value is preferred. The gray background represents the results with LCGen.

| Metrics | Dreamfusion | | Magic3D | | Prolificdreamer | |
| | Origin | LCGen | Origin | LCGen | Origin | LCGen |
| --- | --- | --- | --- | --- | --- | --- |
| JR (%) ↓ | 56.67 | 16.67 | 46.67 | 23.33 | 63.33 | 20.00 |
| CS (%) ↑ | 22.73 | 22.95 | 23.77 | 23.61 | 26.23 | 28.94 |

Janus Problem occurring. Meanwhile, the CS reflects that LCGen improves spatial consistency without compromising the quality of the generated images. This demonstrates the effectiveness and practicality of LCGen.

**LCGen vs. Other Methods addressing Janus Problem.** Some current work is also designed to address the Janus Problem [15, 19–21, 28, 40, 13, 1, 12]. However, these methods either require extensive multi-stage fine-tuning or object-specific designs and do not address the fundamental causes of the Janus Problem, as shown in Fig. 6 (See more comparison in Appendix G). Compared to these methods, our approach has the following advantages: 1) It targets the essence of the Janus Problem by tapping into the multi-perspective information within 2D priors without the need to introduce additional information. 2) It can be directly integrated into any SDS-based text-to-3D method. 3) There is no need to alter the training paradigm, and the computational cost for certainty calculations is negligible compared to the baseline.

Figure 6: Comparison of different methods dealing with Janus Problem.

| Method | No Additional Prior | Single Stage and No Fine-tuning | No Object-specificity | JR (%) | CS (%) |
|---|---|---|---|---|---|
| MVDream [28] | ✗ (3D data) | ✗ | ✓ | 20.00 | 26.17 |
| Prep-Neg [1] | ✓ | ✓ | ✗ | 26.67 | 26.23 |
| D-SDS [12] | ✗ (LLM) | ✓ | ✓ | 23.33 | 24.82 |
| DreamControl [13] | ✓ | ✗ | ✓ | 20.00 | 28.14 |
| LCGen (Prolificdreamer) | ✓ | ✓ | ✓ | 20.00 | 28.94 |

## 5.3 Ablation Study

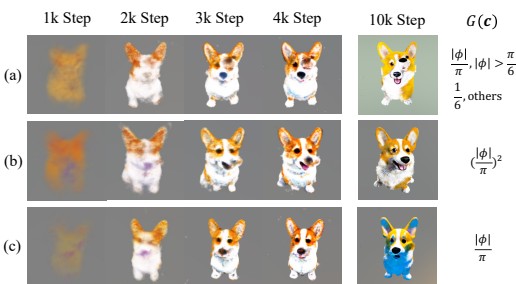

Figure 7: The impact of different choices of $G(c)$.

**Ablation of $G(c)$.** $G(c)$ is an important hyperparameter in LCGen, representing the preference for the selection tendency of generation certainty. The larger the $G(c)$, the higher the suppression of certainty, and the more inclined it is towards low certainty generation. For the text prompt "a corgi" in ProlificDreamer, we designed different $G(c)$ as shown in Fig. 7. It can be observed that when $G$ is a piecewise function of $|\phi|$ in Fig. 7(a), there is a possibility of generating multiple faces from different front views; when using an absolute value function of $\phi$, it can generate correctly but also exhibits subtle differences in the generation process, as shown in Fig. 7(b)(c). See Appendix. E for $G$ selection.

**Certainty within the training step windows.** We also conducted ablation experiments on the certainty at different time step windows during the training process. As shown in Fig. 8, for each step, a window $(step - 200, step]$ is selected, and the variance of certainty within the window is calculated. By subtracting the window variance values of the LCGen and origin methods, the results are obtained. It can be observed that during the training process, the certainty variance of most step windows in LCGen is larger. This indicates that the LCGen method effectively separates the certainty of different viewpoints.

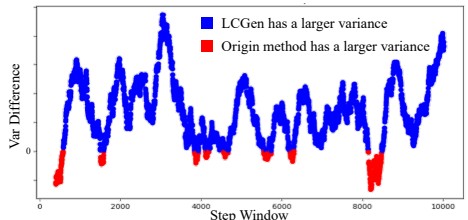

Figure 8: The difference in the variance of certainty within the training step windows between w/ and w/o LCGen.

## 5.4 Visualization

In this Section, we conduct a visualization analysis of the LCGen and the corresponding certainty for each viewpoint. Our chosen textual example is "A corgi". We use ProlificDreamer [34] as our baseline and achieve results after a maximum of 10,000 steps.

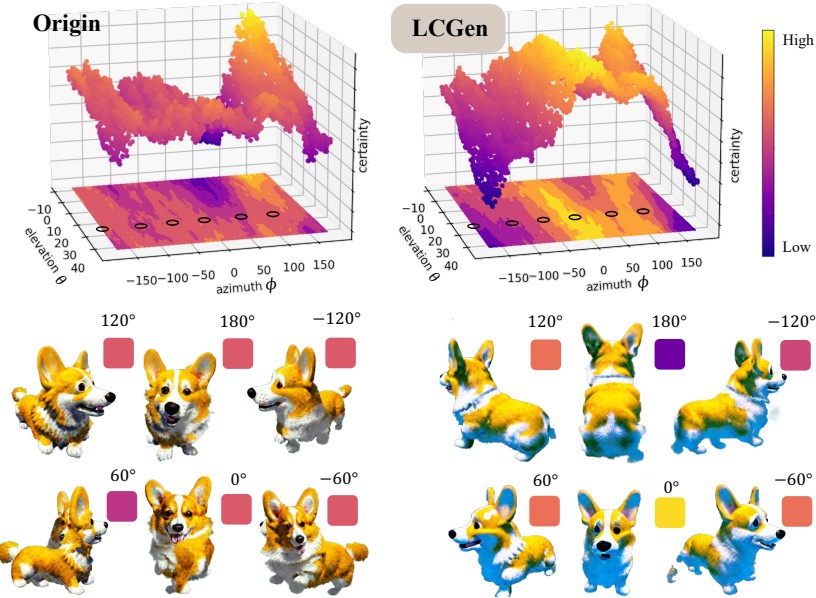

Figure 9: Visualization of Certainty after Generation w/o and w/ LCGen. The upper half of the 3D scatter plot has the camera view parameters $(\theta, \phi)$ on the x and y axes, and Certainty on the z-axis. The lower half depicts the generation results from different views, along with their $\phi$ and the Certainty colors as represented in the scatter plot.

As shown in Fig. 9, the left side shows the original ProlificDreamer results, while the right side features the ProlificDreamer using the LCGen method. For each method, the lower half presents visualized outputs showing rendering results at various azimuths $\phi$ with an elevation $\theta$ of $-15°$. It can be observed that the original prolificdreamer generates the corgi with faces on both the front and back, which is indicative of the Janus Problem. In contrast, the 3D representation created using LCGen exhibits high spatial consistency. We have also visualized the certainty obtained from the diffusion model after inputting images of the fully trained NeRF from various viewpoints. It is shown that due to the presence of multiple faces, the certainty in the original method does not show a clear pattern of change, consistent with the analysis presented earlier. In comparison, LCGen demonstrates a more distinct pattern of certainty varying with $c$ where the certainty is greater when $|\phi|$ is small and decreases as $|\phi|$ increases. This highlights the effectiveness of LCGen's constraints on certainty and also corroborates the relationship between certainty and $c$. See details in Appendix E.

## 6 Related Works

**Methods addressing Janus Problem.** Past research has explored multi-stage networks that utilize 3D priors to reduce the Janus Problem. A two-stage 2D lifting framework has been proposed in DreamControl [13], leveraging 3D self-priors to enhance geometric consistency in 3D generation. Other approaches, such as Perp-Neg [1], innovate by using negative prompts in diffusion models to remove undesirable attributes or views while maintaining the core concept. However, these approaches either necessitate extensive multi-stage pre-training or are tailored to specific objects, without tackling the root causes of the Janus Problem. Additionally, they often require specialized and resource-intensive procedures. See details in Appendix. G

## 7 Conclusion and Discussion

In this study, we initially model the Janus Problem and analyze its causes visually. We then introduce LCGen to guide text-to-3D generation toward spatial consistency by establishing varied certainty priors across viewpoints. Our method, validated through experiments, can integrate seamlessly with

various SDS-based text-to-3D methods to mitigate the Janus Problem. It does so without adding extra data requirements, excessive computational overhead, or degrading the quality of generated outputs.

**Limitations and Broader Impacts.** Our method performs well in generating individual objects but has limitations with complex multi-object scenes. Due to the lack of 3D training data resulting in an insufficient understanding of the 3D world, LCGen still has some failure cases in the text-to-3D Janus Problem. Additionally, using a fixed $G$ leaves room for improvement in adaptive generation. At the same time, realistic AI-generated content may have adverse social impacts. Like other generative model researchers, we must remain vigilant and take precautions against generating false content.

**Acknowledgements.** This work was supported by the National Natural Science Foundation of China under Grant 62406075, National Key Research and Development Program of China under Grant 2023YFC3604802; in part by the China Postdoctoral Science Foundation under Grant 2023M730647 and Grant 2023TQ0075.

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

# A  Symbol Reference Table

Table 1: Symbol Reference Table

| Symbol | Meaning | Description |
|---|---|---|
| $\boldsymbol{x}_t$ | State variable or rendered image | Represents the state at time $t$ in the diffusion process or rendered image from 3D models |
| $q(\boldsymbol{x}_t\|\boldsymbol{x}_{t-1})$ | Transition probability distribution | Probability of state $\boldsymbol{x}_{t-1}$ given state $\boldsymbol{x}_t$ during the diffusion forward process |
| $p(\boldsymbol{x}_{t-1}\|\boldsymbol{x}_t)$ | Transition probability distribution | Probability of state $\boldsymbol{x}_{t-1}$ given state $\boldsymbol{x}_t$ during the diffusion reverse process |
| $\beta_t$ | Variance parameter | Variance term at time $t$ in the diffusion process |
| $\mathbf{I}$ | Identity matrix | Identity matrix with ones on the diagonal and zeros elsewhere |
| $\mathcal{N}$ | Gaussian distribution | Normal (Gaussian) distribution |
| $\mu, \boldsymbol{\mu}$ | Mean | Mean of the Gaussian distribution |
| $\sigma^2, \Sigma, \boldsymbol{\Sigma}$ | Variance & covariance matrix | Variance & covariance matrix of the Gaussian distribution |
| $\Phi$ | Pretrained model | Pretrained 2D diffusion model |
| $\Theta$ | 3D model | 3D model to be optimized |
| $\mathcal{L}_{\text{SDS}}$ | SDS loss function | Loss function used in Score Distillation Sampling |
| $\mathcal{L}_{\text{VSD}}$ | VSD loss function | Loss function used in Variational Score Distillation |
| $\Omega$ | Weighting factor | Weighting factor used in $\mathcal{L}_{\text{SDS}}$ and $\mathcal{L}_{\text{VSD}}$ |
| $\epsilon$ | Noise | Noise term in the diffusion process |
| $\hat{\epsilon}_t$ | Predicted noise | Noise predicted by the diffusion model at time $t$ |
| $t$ | Time step | Time step in the diffusion process |
| $y$ | Text prompt | The text input that guides the generation process in text-to-3D or text-to-image models |
| $c, \boldsymbol{c}$ | Camera viewpoint parameter | Represents camera viewpoint parameters in the 3D model |
| $b, \boldsymbol{b}$ | Head position in 3D representation | Represents the position of the head in the 3D model |
| $S^2$ | Sphere | The 2D spherical surface on which $\boldsymbol{c}$ and $\boldsymbol{b}$ lie |
| $\mathcal{I}$ | Interval | Regions of the sphere $S^2$ |
| $w_i$ | Weight of the $i$-th Gaussian component | Represents the weighting factor for the $i$-th Gaussian component in a mixture model |
| $p(\boldsymbol{b}, \boldsymbol{c})$ | Probability | Probability of generating a head at position $\boldsymbol{b}$ under viewpoint $\boldsymbol{c}$ |
| $P(\boldsymbol{b})$ | Probability | Probability of the head appearing at position $\boldsymbol{b}$ |
| $A$ | Solid angle element | Differential solid angle element in spherical coordinates |
| $(\theta, \phi)$ | Spherical coordinates | The elevation and azimuth used to represent points on $S^2$ |
| $\alpha_{t-1}$ | Coefficient | Coefficient related to $\beta_t$ in the diffusion process |
| $C^c(\boldsymbol{x}_{t-1}\|\boldsymbol{x}_t)$ | Certainty | Defined certainty in LCGen |
| $\mathcal{L}_{\text{cert}}$ | Certainty loss function | Loss function used in LCGen to ensure certainty consistency |
| $\gamma$ | Normalization constant | Constant used for normalization in $\mathcal{L}_{\text{cert}}$ |
| $G(\boldsymbol{c})$ | View-based guidance | Function for view-based guidance in $\mathcal{L}_{\text{cert}}$ |
| $\omega$ | Weighting factor | Weighting factor used in $\mathcal{L}_{\text{cert}}$ |

# B  View Distribution in 2D Diffusion Model

We generated 100 2D images for each text condition $y$ of $\mathcal{I}$ in Stable Diffusion 2.1 base [27], with the texts $y$ being "a pig wearing a backpack, front view", "a pig wearing a backpack, side view",

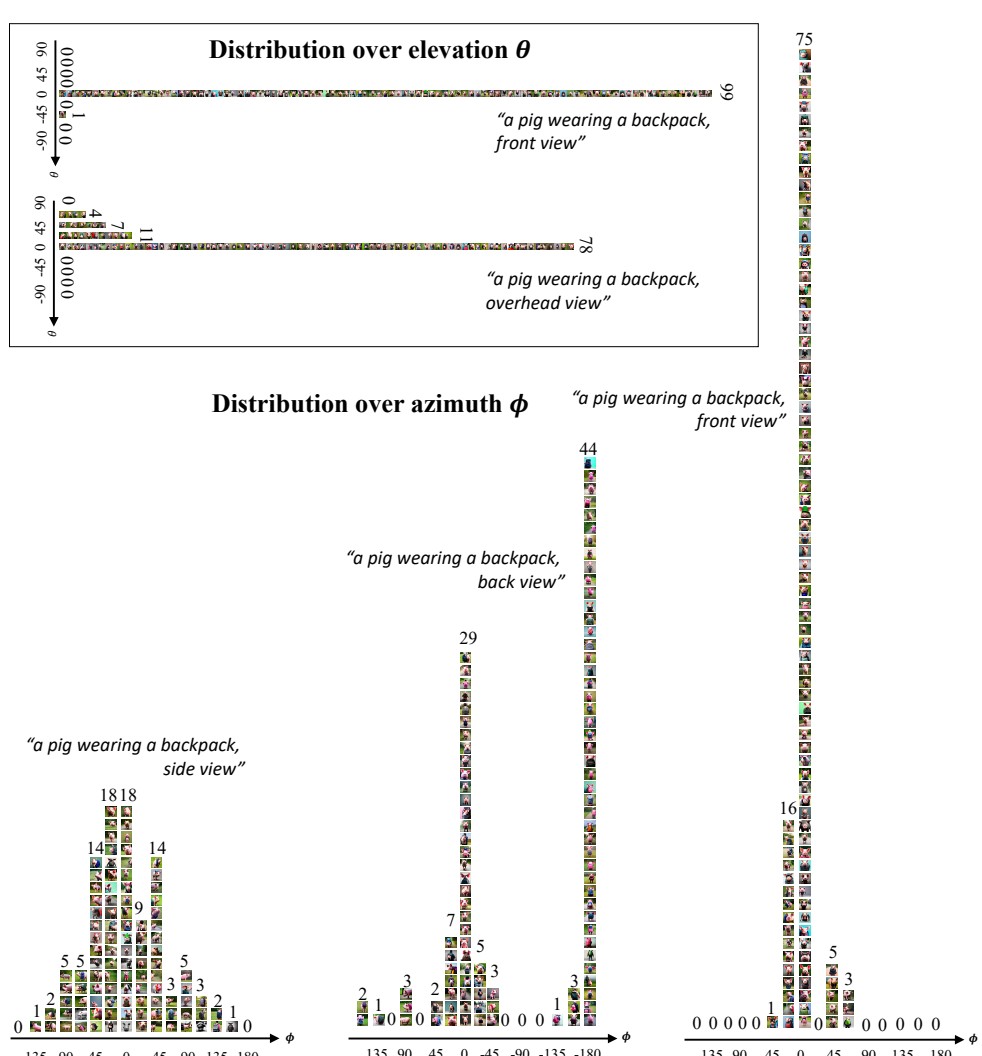

Figure 10: View distributions given different text conditions in 2D diffusion model.

"a pig wearing a backpack, back view", and "a pig wearing a backpack, overhead view". For the generated images, we manually assessed and categorized the camera parameters in the elevation $\theta$ and azimuth $\phi$ dimensions, obtaining the results shown in Fig. 10.

It can be observed that the images generated by the 2D diffusion model have some variation in view distribution corresponding to different $y$ of $\mathcal{I}$, exhibiting different peaks. From the $\theta$ dimension, in the front view, almost all (99%) of the generated 2D images are at $\theta = 0$. If the text is changed to an overhead view, 22% of the images are above $\theta = 0$. In the azimuth dimension, when the view is front, 75% of the generated images are concentrated at $\phi = 0$; when the view is back, 44% of the images are at $\phi = 180$, while 29% are at $\phi = 0$; for the side view, the distribution is relatively scattered, generally symmetrically distributed around $\phi = 0$, exhibiting diverse view characteristics.

This corresponds to the content in Sec. 3. When different $c$ within $\mathcal{I}$ are given the same $y$, the 2D diffusion guidance follows the same distribution. In this work, we model this distribution as a combination of Gaussian distributions, as shown in Eq. 5.

## C  Detailed Analysis of Janus Problem

According to Sec. 3, SDS-based text-to-3D methods utilize a view-dependent prompt approach with discrete encoding. In the context of the sphere $S^2$, based on elevation and azimuth, Camera parameters $c$ are classified into intervals $\mathcal{I}$ representing four view categories: side, front, back, and overhead, as shown in Fig. 2 (a).

However, text prompts are generally limited to high-level descriptions and do not offer the precise control achievable in actual photography. This issue largely stems from the lack of detailed data in diffusion training datasets, where specific camera settings are rarely documented [7]. As a result, images generated based on viewpoint text conditions in the diffusion model tend to be imprecise and show significant viewpoint fluctuations. Images of the front may be generated under guidance from textual prompts of different viewpoints [1]. To address this, the conditional probability density function $p_t^{\mathcal{I}}(\boldsymbol{x}_t \mid c, y)$ for each interval $\mathcal{I}$ is modeled as a mixture of multivariate Gaussian distributions, each parameterized by $N_i^{\mathcal{I}}(\boldsymbol{\mu}_i^{\mathcal{I}}, \boldsymbol{\Sigma}_i^{\mathcal{I}})$.

Specifically, taking a one-dimensional Gaussian distribution as an example, for camera parameter $c \in \mathcal{I}$, the probability density for synthesizing a head at a position with bias $b \in S^2$ is denoted as follows:

$$p^{\mathcal{I}}(b, c) = \sum_i w_i^{\mathcal{I}} \frac{1}{\sqrt{2\pi\sigma_i^{\mathcal{I}^2}}} \exp\left(-\frac{(c - b - \mu_i^{\mathcal{I}})^2}{2\sigma_i^{\mathcal{I}^2}}\right) \tag{14}$$

where: - $w_i^{\mathcal{I}}$ are the mixture weights for each Gaussian component in interval $\mathcal{I}$, - $\mu_i^{\mathcal{I}}$ and $\sigma_i^{\mathcal{I}^2}$ are the mean and variance of the $i$-th Gaussian component, - $\sum_i w_i^{\mathcal{I}} = 1$ to ensure that the total probability of the mixture distribution integrates to one over its domain.

As shown in Fig. 2 (b), the integral of $p^{\mathcal{I}}$ with respect to $c$ over the interval $\mathcal{I}$ is given by:

$$P^{\mathcal{I}}(b) = \int_{\mathcal{I}} p^{\mathcal{I}}(b, c) \, dc = \sum_i w_i^{\mathcal{I}} \int_{c_0}^{c_1} \frac{1}{\sqrt{2\pi\sigma_i^{\mathcal{I}^2}}} \exp\left(-\frac{(c - b - \mu_i^{\mathcal{I}})^2}{2\sigma_i^{\mathcal{I}^2}}\right) \, dc. \tag{15}$$

Here, $\mathcal{I} = [c_0, c_1]$ represents the view interval of integration. The integral $P^{\mathcal{I}}$ provides the probability density function $PDF$ of Janus Problem occurring at $b$.

We now extend our analysis to the multivariate Gaussian distribution, as shown in Fig. 2 (c). This extension actually incorporates several inherent settings regarding camera geometry in 3D space, as shown in Fig. 11.

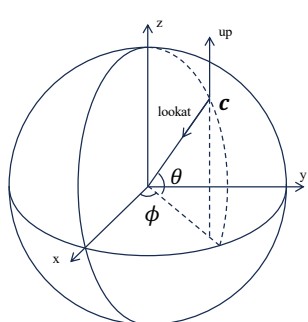

Figure 11: Camera geometry in 3D space.

1. The camera's "lookat" axis intersects the world coordinate origin, effectively fixing two rotational degrees of freedom of $\boldsymbol{c}$,

2. The "up" axis is aligned with the z-axis of the world coordinate system, thereby constraining another rotational degree of freedom,

3. The assumptions hold under a constant distance between the camera and the world coordinate origin, which fixes the translational freedom along the radial axis from the origin—a parameter that does not explicitly influence the perspective modeling.

Given these conditions, the remaining camera parameters $\boldsymbol{c}$ can be described by two active degrees of freedom: elevation $\theta$ and azimuth $\phi$. If we evaluate the impact of the vector $\boldsymbol{b}$ on this distribution over a specified interval $\mathcal{I}$, the resulting integral $P^{\mathcal{I}}(\boldsymbol{b})$ can be interpreted as the probability density of $\boldsymbol{b}$ over that interval:

$$P^{\mathcal{I}}(\boldsymbol{b}) = \int_{\mathcal{I}} p^{\mathcal{I}}(\boldsymbol{b}, \boldsymbol{c})\, d\boldsymbol{c}$$

$$= \sum_i w_i^{\mathcal{I}} \int_{\mathcal{I}} \frac{1}{\sqrt{(2\pi)^k |\boldsymbol{\Sigma}_i^{\mathcal{I}}|}} \exp\left( -\frac{1}{2}(\boldsymbol{c} - \boldsymbol{b} - \boldsymbol{\mu}_i^{\mathcal{I}})^\top \boldsymbol{\Sigma}_i^{\mathcal{I}^{-1}}(\boldsymbol{c} - \boldsymbol{b} - \boldsymbol{\mu}_i^{\mathcal{I}}) \right) d\boldsymbol{c}. \tag{16}$$

Here, $k = 2$ is the dimension of the space, $\boldsymbol{c} \in \mathcal{I}$ and $\boldsymbol{b} \in S^2$ are vectors defining the position in this space, and $\boldsymbol{\mu}_i^{\mathcal{I}}$ and $\boldsymbol{\Sigma}_i^{\mathcal{I}}$ represent the mean vector and covariance matrix of the distribution, respectively.

Recognizing that different intervals $\mathcal{I}$ yield distinct probability functions, the total probability density $P(\boldsymbol{b})$ for the Janus Problem occurring at the position defined by $\boldsymbol{b} \in S^2$ is calculated as the sum of integrals over distinct regions $\mathcal{I}$ of the sphere:

$$P(\boldsymbol{b}) = \sum_{\mathcal{I}} \int_{\mathcal{I}} p^{\mathcal{I}}(\boldsymbol{b}, \boldsymbol{c})\, dA = \sum_{\mathcal{I}} \int_{\mathcal{I}} p^{\mathcal{I}}(\boldsymbol{b}, \theta, \phi) \sin(\theta)\, d\theta\, d\phi \tag{17}$$

where $\mathcal{I}$ represents different regions of the sphere, $dA$ is the differential solid angle element in spherical coordinates, and $\sin(\theta)$ accounts for the area element on the sphere. Each integral $\int_{\mathcal{I}}$ calculates the contribution to $P(\boldsymbol{b})$ from each region $\mathcal{I}$.

While direct integration is theoretically feasible, the integration of multivariate Gaussian distributions can present considerable complexity in practical execution. Consequently, numerical integration techniques are employed to approximate $P(\boldsymbol{b})$ effectively, as shown in Fig. 2 (d). It can be observed that when the parameters $\mathcal{I}$, $N_i^{\mathcal{I}}(\boldsymbol{\mu}_i^{\mathcal{I}}, \boldsymbol{\Sigma}_i^{\mathcal{I}})$, and $w_i^{\mathcal{I}}$ are altered, $P(\boldsymbol{b})$ exhibits a high likelihood of manifesting the Janus Problem at various points $\boldsymbol{b} \in S^2$.

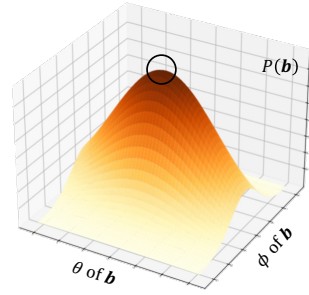

This provides us with an approach to address the Janus Problem. If we can decouple the relationship between viewpoints and distributions, allowing different $\boldsymbol{c}$ to follow different distributions (with $\boldsymbol{\mu}$ coinciding with $\boldsymbol{c}$), we can achieve the modeling results shown in Fig. 12. It can be observed that there is only one probability peak in $S^2$, indicating the presence of only one head. Based on this, we propose LCGen, utilizing the method of low certainty generation, to effectively mitigate the Janus Problem, as shown in Sec. 4.

Figure 12: Modeling result without Janus Problem.

## D Back Propagation Details

For $\mathcal{L}_{\text{cert}}$, we obtain:

$$\frac{\partial \mathcal{L}_{\text{cert}}}{\partial \Theta} = \frac{\partial \mathcal{L}_{\text{cert}}}{\partial \boldsymbol{x}_t} \cdot \frac{\partial \boldsymbol{x}_t}{\partial \Theta} \tag{18}$$

Using the chain rule, we first calculate the partial derivative with respect to $x_t$, involving derivatives of $\boldsymbol{\mu}^{\boldsymbol{c}}$ and $\boldsymbol{\sigma}^{\boldsymbol{c}}$. Given $\boldsymbol{\mu}^{\boldsymbol{c}}$ and $\boldsymbol{\sigma}^{\boldsymbol{c}}$, since $\boldsymbol{\sigma}^{\boldsymbol{c}}$ does not depend on $\boldsymbol{x}_t$, its derivative is zero. The derivative of $\boldsymbol{\mu}^{\boldsymbol{c}}$ is:

$$\frac{\partial \boldsymbol{\mu}^{\boldsymbol{c}}}{\partial \boldsymbol{x}_t} = \frac{1}{\sqrt{1 - \beta_t}} \tag{19}$$

Since $\boldsymbol{\sigma}^{\boldsymbol{c}}$ is independent of $\boldsymbol{x}_t$, inserting into the expression for $C^{\boldsymbol{c}}(\boldsymbol{x}_{t-1}|\boldsymbol{x}_t)$ and using properties of the Gaussian distribution, we have:

$$\frac{\partial p^{\boldsymbol{c}}}{\partial \boldsymbol{x}_t} = -\frac{(\boldsymbol{x}_{t-1} - \boldsymbol{\mu}^{\boldsymbol{c}})}{\boldsymbol{\sigma}^{\boldsymbol{c}2}} \cdot C^{\boldsymbol{c}}(\boldsymbol{x}_{t-1}|\boldsymbol{x}_t) \cdot \frac{\partial \boldsymbol{\mu}^{\boldsymbol{c}}}{\partial \boldsymbol{x}_t} \tag{20}$$

Since $G(\boldsymbol{c})$ is independent of $\boldsymbol{x}_t$, incorporating the derivative into $\frac{\partial \mathcal{L}_{\text{cert}}}{\partial \boldsymbol{x}_t}$:

$$\frac{\partial \mathcal{L}_{\text{cert}}}{\partial \boldsymbol{x}_t} = \frac{1}{\gamma} \frac{\partial p^{\boldsymbol{c}}}{\partial \boldsymbol{x}_t} G(\boldsymbol{c}) = -\frac{1}{\gamma} C^{\boldsymbol{c}}(\boldsymbol{x}_{t-1}|\boldsymbol{x}_t) G(\boldsymbol{c}) \cdot \frac{(\boldsymbol{x}_{t-1} - \boldsymbol{\mu}^{\boldsymbol{c}})}{\boldsymbol{\sigma}^{\boldsymbol{c}2}} \cdot \frac{\partial \boldsymbol{\mu}^{\boldsymbol{c}}}{\partial \boldsymbol{x}_t} \tag{21}$$

Substituting the expression for $\boldsymbol{\mu}^{\boldsymbol{c}}$, we get:

$$\nabla_{\Theta} \mathcal{L}_{\text{Cert}}(\Theta) = \mathbb{E}_{t,\boldsymbol{\epsilon},\boldsymbol{c}} \left[ \omega(t) \cdot \frac{1}{\gamma} \cdot C^{\boldsymbol{c}}(\boldsymbol{x}_{t-1}|\boldsymbol{x}_t) \cdot G(\boldsymbol{c}) \cdot \frac{\partial \boldsymbol{x}_t}{\partial \Theta} \right]$$

$$= \mathbb{E}_{t,\boldsymbol{\epsilon},\boldsymbol{c}} \left[ \omega(t) \cdot \mathcal{L}_{\text{cert}} \cdot \frac{\partial \boldsymbol{x}_t}{\partial \Theta} \right] \tag{22}$$

where

$$\omega(t) = -\frac{\boldsymbol{x}_{t-1} - \frac{1}{\sqrt{1-\beta_t}}(\boldsymbol{x}_t - \sqrt{1-\beta_t}\hat{\boldsymbol{\epsilon}}_t)}{\boldsymbol{\sigma}^{\boldsymbol{c}2}} \cdot \frac{1}{\sqrt{1-\beta_t}} \tag{23}$$

# E   Implementation Details

## E.1   Analysis of Janus Problem

In the Sec. 3 and C, we simulated different data distributions by setting different values of $\boldsymbol{\mu}$ and $\boldsymbol{\Sigma}$ in Eq. 16. To simplify the model, we use $\sigma^2$ instead of $\boldsymbol{\Sigma}$.

**Intervals and Boundaries**

1. Threshold across elevation $\theta$:
$$t_{\text{overhead}} = \frac{\pi}{4}$$

2. Thresholds across azimuth $\phi$:
$$t_{\text{front}} = \frac{\pi}{6}, \quad t_{\text{back}} = \frac{\pi}{8}$$

**Parameters of Gaussian Distributions**

1. Side view:

- Integration Range of $\mathcal{I}_{\text{side}}$:

$$\begin{cases} 0 \le \theta < \pi - t_{\text{overhead}} \\ \phi < \frac{\pi}{2} - t_{\text{front}} \text{ or } \phi > \frac{3\pi}{2} + t_{\text{back}} \end{cases} \quad \text{and} \quad \begin{cases} 0 \le \theta < \pi - t_{\text{overhead}} \\ \frac{\pi}{2} + t_{\text{front}} < \phi < \frac{3\pi}{2} - t_{\text{back}} \end{cases}$$

- Parameters of Gaussian Distribution 1:
$$(\theta, \phi) = \left( \frac{\pi}{2}, b_\phi \right), \quad \sigma_1 = 0.3$$

- Parameters of Gaussian Distribution 2:
$$(\theta, \phi) = \left( \frac{\pi}{2} + b_\theta, \frac{\pi}{2} + b_\phi \right), \quad \sigma_2 = 0.3$$

2. Front view:

- Integration Range of $\mathcal{I}_{\text{front}}$:

$$\begin{cases} 0 \le \theta < \pi - t_{\text{overhead}} \\ \frac{\pi}{2} - t_{\text{front}} \le \phi \le \frac{\pi}{2} + t_{\text{front}} \end{cases}$$

- Parameters of Gaussian Distribution 3:
$$(\theta, \phi) = \left( \frac{\pi}{2}, \pi + b_\phi \right), \quad \sigma_3 = 0.2$$

3. Back view:

- Integration Range of $\mathcal{I}_{\text{back}}$:

$$\begin{cases} 0 \leq \theta < \pi - t_{\text{overhead}} \\ \frac{3\pi}{2} - t_{\text{back}} \leq \phi \leq \frac{3\pi}{2} + t_{\text{back}} \end{cases}$$

- Parameters of Gaussian Distribution 4:

$$(\theta, \phi) = \left( \frac{\pi}{2} - b_\theta, \frac{3\pi}{2} + b_\phi \right), \quad \sigma_4 = 0.4$$

4. Overhead view:

- Integration Range of $\mathcal{I}_{\text{overhead}}$:

$$\theta \geq \pi - t_{\text{overhead}}$$

- Parameters of Gaussian Distribution 5:

$$(\theta, \phi) = \left( \pi - |b_\theta|, \frac{\pi}{2} + b_\phi \right), \quad \sigma_5 = 0.3$$

**Integral Computation Range of $b$**

1. $b_\theta$ Values:

$$b_\theta \text{ ranges from } -\frac{\pi}{2} \text{ to } \frac{\pi}{2} \text{ with 50 intervals}$$

2. $b_\phi$ Values:

$$b_\phi \text{ ranges from } -\pi \text{ to } \pi \text{ with 50 intervals}$$

From this, we can obtain the 3D plot in Fig. 2(d). By varying $\mathcal{I}$ and the parameters of the Gaussian distribution, we can model the probability of generating a Janus Problem at different $b$ under different distributions. The location of the probability peak represents the potential position where a head might appear.

### E.2 LCGen.

We implement original methods and LCGen based on threestudio [8] and a single A100 GPU, using the PyTorch framework. For Dreamfusion, Magic3D, and ProlificDreamer, the hyperparameters are all kept consistent with the default settings in config files in the repository. Specifically, according to the instructions in threestudio,

- In Dreamfusion, to prevent scene being stuffed with floaters/becoming empty, we set system.loss.lambda_sparsity=0.1.

- In Dreamfusion and Magic3D, to prevent the model incorrectly treating the background as part of the object, we replace the background with random colors with a probability 0.5 by setting system.background.random_aug=true.

In the experiment, we set $G(c) = |\phi|$, and obtained the results after a maximum of 10,000 steps. We also attempted to embed $\phi$ into $G$. In the experiments, we found that the multi-head problem almost never appeared in the $\theta$ dimension, and setting $\theta$ did not significantly affect the results. Therefore, we ultimately chose the above $G$. For the sake of experimental consistency, we have chosen the Stable Diffusion 2.1 base [27] as guidance and NeRF [24] as the 3D representation in the SDS-based method. In $\mathcal{L}_{\text{cert}}$, we use $\gamma$ as the normalization parameter. In our experiments, we set $\gamma$ to 10, which provides the most stable mitigation of the Janus problem.

### E.3 Visualization.

In Sec. 5.4, to make the results clearer, we smooth the values at each position. Specifically, we take the average certainty of all points within a distance of 0.1 from the given position and use this average as the certainty of the scatter point. This helps to avoid disturbances in the visualization results caused by factors such as random view sampling and $t$.

## F    Metrics

In the text-to-3D task, it is difficult to quantitatively evaluate the generated results. This task lacks ground truth for objective scoring. In this work, our goal is to reduce the occurrence of the Janus Problem, so it is necessary to assess the 3D consistency of the generated models and the extent to which the Janus Problem is mitigated. Similar to DreamControl [13], we choose CLIP-Score (CS) and Janus Problem Rate (JR) as our quantitative metrics in Sec. 5.

**CLIP-Score (CS).** The CLIP Score [41] is calculated using the CLIP (Contrastive Language-Image Pretraining) model [26], which is a language-image model learned through contrastive learning. The CLIP model consists of an image encoder and a text encoder, enabling it to measure the similarity between different modalities. The CLIP Score measures the Cosine Similarity between two embedded features. This implementation utilizes the pretrained CLIP model to calculate the mean average of cosine similarities between text and generated images from different views.

**Janus Problem Rate (JR).** To assess the consistency of 3D geometries, we count the instances of inconsistent 3D content produced by each method and calculate their proportion in all content. This represents the occurrence rate of the Janus problem (JR).

## G    Comparison with other methods

Table 2: Comparison of different methods dealing with Janus Problem.

| Method | Task | No Additional Prior | Single Stage | No Fine-tuning | No Object-specificity |
|---|---|---|---|---|---|
| SweetDreamer [15] | Image-to-3D | ✗ (3D data) | ✗ | ✗ | ✓ |
| SyncDreamer [20] | Image-to-3D | ✗ (3D data) | ✗ | ✗ | ✓ |
| Wonder3D [21] | Image-to-3D | ✗ (3D data) | ✗ | ✗ | ✓ |
| EfficientDreamer [40] | Image-to-3D | ✗ (3D data) | ✗ | ✗ | ✓ |
| Zero-1-to-3 [19] | Image-to-3D | ✗ (3D data) | ✗ | ✗ | ✓ |
| MVDream [28] | Text-to-3D | ✗ (3D data) | ✗ | ✗ | ✓ |
| Prep-Neg [1] | Text-to-3D | ✓ | ✓ | ✓ | ✗ |
| D-SDS [12] | Text-to-3D | ✗ (LLM) | ✓ | ✓ | ✓ |
| DreamControl [13] | Text-to-3D | ✓ | ✗ | ✗ | ✓ |
| LCGen (Ours) | Text-to-3D | ✓ | ✓ | ✓ | ✓ |

Table. 2 presents a comparison of our LCGen method with other approaches addressing the Janus Problem. It can be observed that some other methods utilize additional prior information, some employ multi-stage training strategies and fine-tuning of pre-trained models, and some require object-specific designs. In contrast, our method does not require any additional information and can be directly embedded into existing SDS-based text-to-3D methods without fine-tuning the models. It alleviates the Janus Problem with minimal computational cost.

## H    Some Other Results

**Examples without concept of heads.** In Fig. 13 A and B, we show the results of 'a street lamp' and 'a tree' as you mentioned. It can be seen that both the original Prolificdreamer and our method achieve spatial consistency generation. Considering that the above two examples do not have obvious differences between different views, we conducted experiments on 'a sunflower' and 'a piano' as shown in Fig.C and D in Fig. 13. It can be seen that the original Prolificdreamer produced spatial inconsistencies, showing multiple frontal images of the flower and multiple keyboards in single piano from different views. Our method successfully alleviated this issue, generating a sunflower and piano with correct front and back images. This indicates that our method is effective not only for objects with heads but also for spatial consistency modeling of other objects.

**Multiple objects with multiple heads.** We will supplement the discussion details in the limitations section of the Main Paper. Multi-object generation is another important and challenging field in

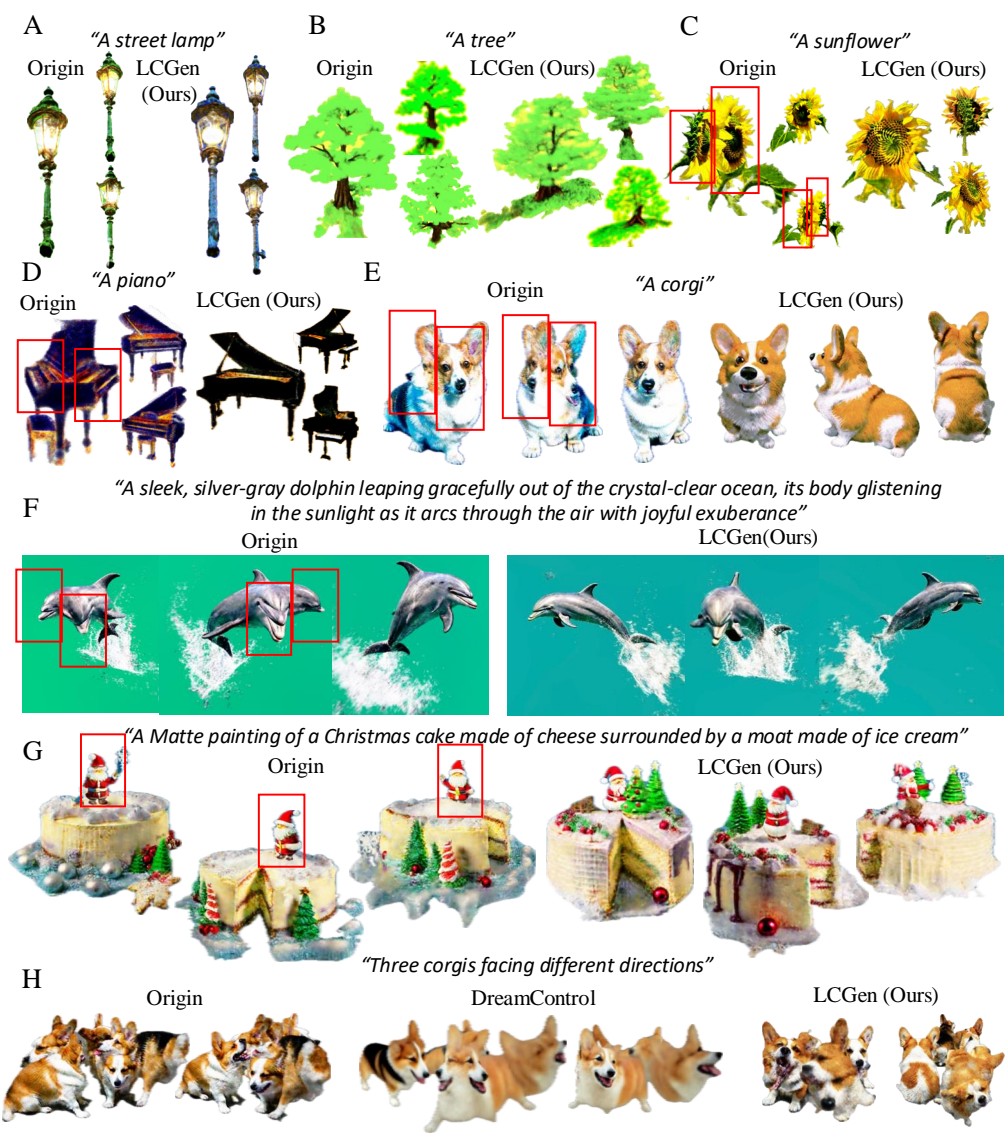

Figure 13: Some other results.

text-to-3D tasks. For SDS-based methods, the current most important issue in multi-object generation is not the Janus problem but how to handle the relationships between different objects. In H in Fig. 13, we show the results of 'three corgis facing different directions.' It can be seen that the original Prolificdreamer cannot correctly generate 3 corgis, and there is some degree of sticking between the objects. Both our method and the current state-of-the-art multi-view consistent generation method DreamControl cannot correctly handle this example. Once the multi-view generation issue is resolved, our work will have more exploration possibilities.

**A large amount of descriptive language.** In Fig. 13 F: "A sleek, silver-gray dolphin leaping gracefully out of the crystal-clear ocean, its body glistening in the sunlight as it arcs through the air with joyful exuberance." It can be observed that the original Prolificdreamer generates two dolphin heads, while our LCGen method correctly generates the image. In Fig. 13 G, we also carefully designed a prompt: "A Matte painting of a Christmas cake made of cheese surrounded by a moat made of ice cream". It can be observed that in the original prolificdreamer, the front and back of Santa Claus on the cake both have faces. However, using our method, the correct samples are generated.

*"an astronaut riding a kangaroo"*

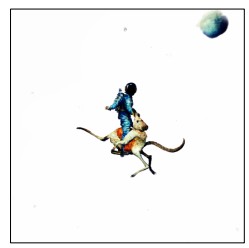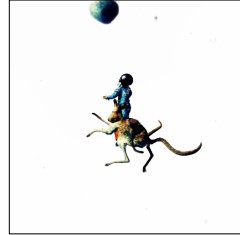

Figure 14: Failure example.

# I Failure Example

Although experiments demonstrate that our method excels in mitigating the Janus Problem, there are still some failure cases. Fig. 14 shows a failure example. By incorporating our method into ProlificDreamer [34] and setting the text prompt $y$ as "an astronaut riding a kangaroo", the method correctly generated a single human head and a single kangaroo head. However, it resulted in issues such as an incorrect human arm position and multiple kangaroo legs. This may be due to the presence of multiple objects in the generated text, making it difficult to separate the combinations of objects from the perspective of certainty.

When objects are in very unusual poses, our method has a certain probability of failing. For example, "an upside-down lion." In this case, using the original certainty to separate different views fails because the bottom might be the highest certainty view. For such samples, we need to redesign the view-based guidance function to accommodate the generation of these special samples.

Our method may fail in multi-object generation scenarios. This is due to the limitations of current text-to-3D methods in multi-object generation tasks. As shown in Fig. 13 H, when the text is "three corgis facing different directions," baseline method Prolificdreamer fails to generate three corgis correctly, with an overlap between different objects. On this basis, the current best view-consistent generation methods, such as DreamControl and our method, cannot directly address the Janus Problem. To solve this issue, the primary task is to solve the multi-object generation problem in text-to-3D, which is a significant research area.

Due to the lack of 3D prior knowledge, like other SDS-based methods, our method can only model 3D representations that appear more realistic from various perspectives, but cannot ensure that these 3D representations adhere to the physical laws of the real world. For example, during the generation of octopus tentacles, since there is no difference in tentacles from different views and the model does not know how many tentacles should be generated, it may produce 3D representations that do not conform to objective reality. To address this issue, we need to endow the model with the ability to understand the 3D world. One possible approach is to collect massive 3D data and design appropriate representation forms to establish a pre-trained 3D generation model (e.g., 3D diffusion). Given the enormous data requirements and training difficulty, this requires the collective effort of the entire AIGC community.

# J Prompt Library

"a bald eagle carved out of wood", "a beagle in a detective's outfit", "a beautiful rainbow fish", "a bichon frise wearing academic regalia", "a cat with a mullet", "a ceramic lion", "a chihuahua wearing a tutu", "a chimpanzee holding a peeled banana", "a chimpanzee looking through a telescope", "a confused beagle sitting at a desk working on homework", "a corgi taking a selfie", "a cute steampunk elephant", "a DSLR photo of a baby dragon drinking boba", "a DSLR photo of a cat wearing a bee costume", "a DSLR photo of a corgi puppy", "a DSLR photo of a dog made out of salad", "a DSLR photo of a frog wearing a sweater", "a DSLR photo of a humanoid robot using a laptop", "a DSLR photo of a lion reading the newspaper", "a DSLR photo of a mouse playing the tuba", "a DSLR

photo of a pig playing a drum set", "a DSLR photo of a robot dinosaur", "a fox playing the cello", "a highland cow", "a lionfish", "a pig wearing a backpack", "a red panda", "a tiger playing the violin", "a zoomed out DSLR photo of a baby dragon", "a zoomed out DSLR photo of a monkey riding a bike"

