# OpenReview forum: "LCGen: Mining in Low-Certainty Generation for View-consistent Text-to-3D"
_NeurIPS.cc/2024/Conference — NeurIPS 2024 poster_

### Official Review · Reviewer_wiP5 · 2024-07-10

**Soundness:** 3
**Presentation:** 3
**Contribution:** 3
**Rating:** 7
**Confidence:** 1

**Summary:**

The paper attempts to address the Janus Problem in SDS-based text-to-3D methods. It first analyzes the cause of the Janus Problem in SDS-based approaches, identifying that discrete view encoding and shared priors in 2D lifting are the primary causes. To address this, it proposes the LCGen method, which guides text-to-3D generation to obtain different priors with varying certainty from different perspectives, thereby ensuring view consistency. Experiments show that the LCGen method can be seamlessly integrated into various SDS-based text-to-3D methods, effectively mitigating the Janus Problem without significant side effects.

**Strengths:**

1. This work attempts to address the challenge of the Janus Problem by tackling its root causes and identifying the key factors that contribute to this issue, making a significant contribution to solving this critical problem.

2. The proposed method addresses the issue from a novel perspective, and experimental results have confirmed its effectiveness.

**Weaknesses:**

1. The examples presented in this work are relatively homogeneous. Have you tried other types of examples? (For instance, a finely designed object)
2. The limitations section and the appendix in the paper show some limitations and failure cases. If this method is supposed to fundamentally address the Janus problem, what are the reasons for these failure cases and limitations? I did not see much detailed discussion about this in the paper.

**Questions:**

See weaknesses

---

> ### Author Rebuttal · Authors · 2024-08-05
>
> Thank you very much for your effort in reviewing our work! Our responses are as follows:
>
> ## 1. Diversified Examples
>
>
> In *Rebuttal File*, we demonstrate how our method alleviates the Janus Problem in various other types of examples.
>
> In **Fig. C and D**, we tested on "a sunflower" and "a piano" to show the effectiveness of our method when the object does not contain a "head". It can be observed that the original Prolificdreamer generates sunflowers and keyboards of piano on both the front and back, resulting in spatial inconsistency. After applying our method, it correctly renders the front and back images of the sunflower and piano, resolving the spatial inconsistency issue.
>
> In **Fig. F**, we carefully designed a prompt to show the effect of our method when a large amount of descriptive language is added: "A sleek, silver-gray dolphin leaping gracefully out of the crystal-clear ocean, its body glistening in the sunlight as it arcs through the air with joyful exuberance." It can be observed that the original Prolificdreamer generates two dolphin heads, while our LCGen method correctly generates the image. In Fig. G, we also carefully designed a prompt: "A Matte painting of a Christmas cake made of cheese surrounded by a moat made of ice cream". It can be observed that in the original prolificdreamer, the front and back of Santa Claus on the cake both have faces. However, using our method, the correct samples are generated.
>
>
>
>
> ## 2. Limitation Discussion
>
> For the failure cases, our analysis is as follows:
>
> 1. When objects are in very unusual poses, our method has a certain probability of failing. For example, "an upside-down lion." In this case, using the original certainty to separate different views fails because the bottom might be the highest certainty view. For such samples, we need to redesign the view-based guidance function to accommodate the generation of these special samples.
>
> 2. Our method may fail in multi-object generation scenarios. This is due to the limitations of current text-to-3D methods in multi-object generation tasks. As shown in Fig. H in *Rebuttal File*, when the text is "three corgis facing different directions," baseline method Prolificdreamer fails to generate three corgis correctly, with an overlap between different objects. On this basis, the current best view-consistent generation methods, such as DreamControl and our method, cannot directly address the Janus Problem. To solve this issue, the primary task is to solve the multi-object generation problem in text-to-3D, which is a significant research area. For example, a current mainstream idea is to generate individual assets for each object and then stitch them together[1]. Our method still has the potential to adapt to such methods.
>
> 3. Due to the lack of 3D prior knowledge, like other SDS-based methods, our method can only model 3D representations that appear more realistic from various perspectives, but cannot ensure that these 3D representations adhere to the physical laws of the real world. For example, during the generation of octopus tentacles, since there is no difference in tentacles from different views and the model does not know how many tentacles should be generated, it may produce 3D representations that do not conform to objective reality. To address this issue, we need to endow the model with the ability to understand the 3D world. One possible approach is to collect massive 3D data and design appropriate representation forms to establish a pre-trained 3D generation model (e.g., 3D diffusion). Given the enormous data requirements and training difficulty, this requires the collective effort of the entire AIGC community.
>
>
> *Thank you again for your efforts and suggestions! We will include detailed discussions on the above points in the Main Paper and Appendix.*
>
> ## Reference
> [1] Zhou, Xiaoyu, et al. "Gala3d: Towards text-to-3d complex scene generation via layout-guided generative gaussian splatting." arXiv preprint arXiv:2402.07207 (2024).

---

### Official Review · Reviewer_FWvW · 2024-07-13

**Soundness:** 3
**Presentation:** 3
**Contribution:** 3
**Rating:** 6
**Confidence:** 2

**Summary:**

This paper presents a simple and effective method to address the Janus Problem for Score Distillation Sampling (SDS)-based text-to-3D methods. This paper argues view consistency is related to that the 3D model tends to learn content with higher certainty from each perspective, and using different priors with different certainty will help with more consistent generation. Specifically, they assume that the certainty of one denoising step follows a Gaussian distribution and design additional loss to constrain the distribution so that different viewpoints have different distributions. Experiments illustrate that their proposed method alleviates the Janus Problem.

**Strengths:**

1. The proposed method is simple and effective. Compared to previous work, this method does not require additional data or models, and can be well integrated into the computation of the original SDS loss.

2. This paper is well-motivated by a detailed analysis of the Janus Problem and the underlying reason for decoupling the relationship between viewpoints and distributions.

3. The empirical experimental results show that it can effectively alleviate the Janus Problem.

**Weaknesses:**

1. One major concern is that this paper lacks enough comparison with previous work. Although in Figure 6, the authors mention several advantages (on the resources, universality, and training efficacy), comparing against the empirical performance quantitatively is still useful. The model may not outperform other models that have more requisites, but it is still important to know where we stand.

2. In addition, for the comparison with DreamControl, the main difference is that this model does not need an additional fine-tuning stage. Because it does not affect the inference time consumption (which is in general more important when we consider the efficacy issue), the contribution will be a bit limited if the proposed method does not outperform DreamControl.

**Questions:**

1. Can you provide a quantitative performance comparison with the previous work on the Janus Problem?

2. Is there a quantitative analysis of the advantages of training time cost compared to DreamControl?

**Limitations:**

The author mentioned the limitation with regard to the modeling of multiple objects or complex objects. Appendix H also includes a failure example of the multiple object issue.

---

> ### Author Rebuttal · Authors · 2024-08-05
>
> Thank you very much for your effort in reviewing our work! Our responses are as follows:
>
> ## 1. Quantitative Comparison with Other Methods Dealing with the Janus Problem
>
> We have provided the quantitative comparison results with other methods in the table in *Rebuttal File* (as well as in the table below). It can be observed that our method can comprehensively surpass or partially match the performance of other methods. Specifically, our method shows the best suppression of the Janus Problem when embedded in dreamfusion; when embedded in prolificdreamer, it achieves the best overall performance, strongly suppressing the Janus Problem while maintaining the highest CLIP score.
>
> Notably, our method can achieve good performance without requiring additional 3D priors or multi-stage fine-tuning, consuming minimal extra computational power (see quantitative comparison in the next response) and can be directly applied to SDS-based text-to-3D methods.
>
>
> | Method           | No Additional Prior | Single Stage | No Fine-tuning | No Object-specificity | JR(%)↓ | CS(%)↑ |
> |------------------|---------------------|--------------|----------------|-----------------------|--------|--------|
> | Zero-1-to-3      | ✘         | ✘            | ✘              | ✔                     | 23.33       | 22.94       |
> | MVDream          | ✘        | ✘            | ✘              | ✔                     |   20.00     | 26.31       |
> | Prep-Neg         | ✔                   | ✔            | ✔              | ✘                     | 26.67       |  26.23      |
> | D-SDS            | ✘             | ✔            | ✔              | ✔                     | 23.33       | 24.82       |
> | DreamControl     | ✔                   | ✘            | ✘              | ✔                     |20.00        |  28.03      |
> | Dreamfusion      |                     |              |                |                       |        |        |
> | Origin           | ✔                   | ✔            | ✔              | ✔                     | 56.67  | 22.73  |
> | LCGen (Ours)     | ✔                   | ✔            | ✔              | ✔                     | **16.67**  | 22.95  |
> | Magic3D          |                     |              |                |                       |        |        |
> | Origin           | ✔                   | ✔            | ✔              | ✔                     | 46.67  | 23.77  |
> | LCGen (Ours)     | ✔                   | ✔            | ✔              | ✔                     | 23.33  | 23.61  |
> | ProlificDreamer  |                     |              |                |                       |        |        |
> | Origin           | ✔                   | ✔            | ✔              | ✔                     | 63.33  | 26.23  |
> | LCGen (Ours)     | ✔                   | ✔            | ✔              | ✔                     | 20.00  | **28.94**  |
>
>
>
> ## 2. Quantitative Comparison of Computation Cost with DreamControl
>
> We conducted a quantitative comparison of computation costs with Dreamcontrol. Notably, in the text-to-3D process, Dreamcontrol consists of two stages: Stage 1 - 3D Self-Prior Generation and Stage 2 - Control-Based Score Distribution. However, only the code for the latter has been released in their official codebase, while the former remains on their to-do list. Using Stage 2 alone cannot directly perform text-to-3D generation when given a text prompt. According to the official instructions, we need to provide an obj file or a threestudio checkpoint as a condition during the inference stage. Therefore, in the quantitative computation process for text-to-3D, we need to consider the costs of both stages (baseline+stage 2 or stage 1+stage 2).
>
> We used prolificdreamer as the baseline model for both methods, since it is one of the best text-to-3D methods currently. We conducted our experiments on a single NVIDIA RTX A6000 GPU with the max steps set to 10,000. **It is worth noting that the following data were all calculated during the text-to-3D generation (i.e., generating the corresponding 3D representation from the given text, not include pre-training time), which can be considered as the inference time in other methods, since the trainable params are the 3d representation we need.** The quantitative comparison of computational overhead in text-to-3D task for one sample is as follows:
>
> | Method           | Trainable params in text-to-3D ↓ | Total estimated model params size ↓ | GPU memory usage ↓ | Text-to-3D Generation Runtime ↓ |
> |------------------|---------------------|--------------|----------------|-----------------------|
> Prolificdreamer (Baseline) | 15.1 M | 60.384 M | 28366 M |  1h30min38s |
> **DreamControl (Stage2)** | *17.6 M* | *70.422 M* | *35554 M* | *1h55min21s*|
> **DreamControl (Baseline + Stage2)** | *32.7 M* | *130.806 M* | *35554 M* | *> 3h25min* |
> **LCGen (Ours)** | *15.1 M* | *60.384 M* | *31458 M* | *1h35min54s* |
>
>
> It can be observed that compared to DreamControl, our method:
>
> 1) Our method does not require two-stage processing, and the model parameter count, and runtime are reduced by more than 50% compared to DreamControl, GPU memory usage is also lower than DreamControl.
>
> 2) When comparing DreamControl (Stage 2) and LCGen (Ours), our method also performs better than DreamControl in all computational overhead metrics.
>
> 3) According to the Table in response 1, the performance of our method is not inferior to DreamControl across all metrics.
>
>
> *Thank you again for your efforts and suggestions! We will include detailed discussions on the above points in the Main Paper and Appendix.*

---

> > ### Comment · Reviewer_FWvW · 2024-08-13
> > **Response to Author's response**
> >
> > I read other reviews and the authors' responses. It seems the authors addressed my issues. Other reviews now also have positive scores.

---

### Official Review · Reviewer_sRbF · 2024-07-13

**Soundness:** 1
**Presentation:** 2
**Contribution:** 2
**Rating:** 5
**Confidence:** 5

**Summary:**

This paper presents a method to tackle the issue of the Janus Problem in text-to-3D content generation method. A method named LCGen has been proposed that focuses on low certainty regions to generate view-consistent generation.

**Strengths:**

Some causes of the Janus Problem have been analysed visually. We then introduce
LCGen method is proposed to guide text-to-3D generation toward spatial consistency by establishing varied certainty priors across viewpoints.
Proposed method works without adding extra data requirements, and excessive computational overhead.

**Weaknesses:**

Proposed method encourage single head, but there could be scenarios where multiple heads needs to be generated e.g.  three corgis facing different directions.
What about those examples where there is no concept of head. How this method will behave in those situations e.g a street lamp, tree etc.
It seems that the proposed low uncertainty approach would lead to low quality samples where fine-grain details are important.
There is no comparison with the existing approaches mitigating Janus problem. A table in the main paper with metrics CS and Janus Rate is a MUST. I acknowledge Table in the appendix G, where only functional level comparison is given. Quantitative comparison is missing.

**Questions:**

Please see the limitations and weakness sections, answer those questions.
Why is there a blue shade in the generated corgi in your method in fig 7 and 10 ?. Is this some kind of artefact due to low uncertainty generation ?

**Limitations:**

It seems the method will work on single object generations. It is also not clear how good this method is compared to the existing method quantitatively.
Examples used for visual validation are also limited corgi and pig, It is not clear which 30 prompts are selected from the library for the experiments. No clarity on the dataset point of view. A clear selection of data samples MUST be added for reproducibility and validation of the proposed method.
Many generated samples in fig 10 are not even of pig.
Only one failure case is shown, while there could be many other scenarios, a detailed analysis of failure cases is also missing. It is important to demonstrate the limitations within which method will work.
While the evaluation is mostly based on the visuals or derived metrics from visuals. The main paper and appendix both doesn’t have sufficient diverse visual results to validate the proposed method

---

> ### Author Rebuttal · Authors · 2024-08-05
>
> Thank you very much for your efforts in reviewing our paper! Below is our response. For Figure and Table, please see *Rebuttal File*.
>
> ## 1. Diverse Examples
>
> The main purpose of our method is to help address Janus Problem appearing on a single object in text-to-3D baselines. In the **limitations section of the *Main Paper*** (lines 291-294), we mention, "Our method performs well in generating individual objects but has limitations with complex multi-object scenes." We will provide more detailed explanations in the limitations section and include the following examples:
>
> 1. **Examples without concept of heads.** In ***Rebuttal File* Fig.A and B**, we show the results of 'a street lamp' and 'a tree' as you mentioned. It can be seen that both the original Prolificdreamer and our method achieve spatial consistency generation. Considering that the above two examples do not have obvious differences between different views, we conducted experiments on 'a sunflower' and 'a piano' as shown in **Fig.C and D in *Rebuttal File***. It can be seen that the original Prolificdreamer produced spatial inconsistencies, showing multiple frontal images of the flower and multiple keyboards in single piano from different views. Our method successfully alleviated this issue, generating a sunflower and piano with correct front and back images. This indicates that our method is effective not only for objects with heads but also for spatial consistency modeling of other objects.
>
> 2. **Multiple objects with multiple heads.** We will supplement the discussion details in the limitations section of the *Main Paper*. Multi-object generation is another important and challenging field in text-to-3D tasks. For SDS-based methods, the current most important issue in multi-object generation is not the Janus problem but how to handle the relationships between different objects. In **Fig.H in *Rebuttal File***, we show the results of 'three corgis facing different directions.' It can be seen that **the original Prolificdreamer cannot correctly generate 3 corgis, and there is some degree of sticking between the objects**. **Both our method and the current state-of-the-art multi-view consistent generation method DreamControl cannot correctly handle this example**. Once the multi-view generation issue is resolved, our work will have more exploration possibilities. For example, a current mainstream idea is to generate individual assets for each object and then stitch them together[1]. **Our method still has the potential to adapt to such methods**.
>
> ## 2. Generation Quality
>
> 1) Regarding the blue shading you mentioned, this is an normal process handled by prolificdreamer during the generation. As shown in **Fig.E in *Rebuttal File***, the original Prolificdreamer can also produce blue shading when changing seed, and our method can generate results without blue shading.
>
> 2) As shown in the *Main Paper*, our method does not suffer from a loss in generation quality compared to baseline methods and can improve the CLIP-Score. Low certainty generation helps the model find an optimization direction that better aligns with specific views, while the fine-grain detail of the generated images is determined by the generation model rather than certainty. Additionally, by aiding in view consistency modeling, the final generation results exhibit spatial consistency and better overall quality.
>
> 3) We also compared some fine-designed prompts, as shown in **Fig.F and G in *Rebuttal File***. Our method ensures high-quality generation even when dealing with finely designed objects.
>
> ## 3. Quantitative Comparison
>
> Please see the **Table in *Rebuttal File*.**
>
> It presents a quantitative comparison with other methods dealing with the Janus Problem. It can be seen that our method, while having a series of advantages, surpasses existing methods in various metrics: it achieves the best overall performance when integrated with Prolificdreamer, strongly suppresses the Janus problem, and maintains the highest CLIP-score; when integrated with Dreamfusion, it has the best suppression effect on the Janus problem.
>
> ## 4. Prompt Library
>
> Due to constraint of 6000 max length, please see **Prompt Library in General Rebuttal to all reviewers**.
>
> We present 30 prompts selected from the Dreamfusion prompt library. Since our focus is on the Janus Problem, we have specifically chosen prompts that can evaluate method's effectiveness.
>
> ## 5. The Pig Generated by Stable Diffusion
>
> All the samples in Fig.10 in *Main Paper* are generated by Stable Diffusion 2.1 base (not our method) to demonstrate that Stable Diffusion 2.1 base has view biases during generation. This supports our analysis and thus introduces our method. In some samples, the pig is not generated correctly, reflecting the limitations of the Stable Diffusion pre-trained model. We generated all the images at once on Stable Diffusion 2.1 base and evaluated their views without any selecting to ensure the accuracy of the presented results.
>
>
> ## 6. Failure Cases
>
> Due to constraint of 6000 max length, please see **Limitation Discussion in respone 2 for reviewer wiP5**. Thanks very much.
>
> *Thank you again for your efforts and suggestions! We will include detailed discussions on the above points in the Main Paper and Appendix.*
>
> ## Reference
> [1] Zhou, Xiaoyu, et al. "Gala3d: Towards text-to-3d complex scene generation via layout-guided generative gaussian splatting." arXiv preprint arXiv:2402.07207 (2024).

---

### Author Rebuttal · Authors · 2024-08-05

Thank you to all the reviewers for their hard work! We are very honored that our work has been recognized for: 1) **significant contribution to addressing key challenges**, 2) **being well-motivated by a detailed analysis of root causes**, 3) **simplicity and effectiveness**, and 4) **good experimental results**.

## Contribution Restatement

Here, we would like to restate our core contributions:

1)	We modeled and analyzed the root causes of the Janus Problem in SDS-based text-to-3D, and designed the LCGen (Low Certainty Generation) method to alleviate the Janus Problem in single-object generation.

2)	Our method can be directly embedded into existing SDS-based text-to-3D methods, effectively alleviating the Janus Problem without compromising generation quality and with minimal computational cost.

## Main Questions

The reviewers' questions mainly focused on two aspects:

1) **Quantitative comparison with other methods dealing with the Janus Problem.** We conducted a quantitative comparison with the methods mentioned in the paper, and the specific results can be found in the table in the rebuttal file. It can be observed that our method outperforms others: it achieves the best overall performance when embedded in prolificdreamer, significantly suppressing the Janus Problem while maintaining the highest CLIP score; our method with dreamfusion achieves the best suppression of the Janus Problem. Additionally, our method does not require the introduction of prior knowledge, multi-stage training, or fine-tuning, and achieves good results with minimal additional computational cost (see response to Reviewer FWvW).

2) **Performance in diversified examples.** Our method aims to alleviate the single-object Janus Problem in SDS-based text-to-3D. We also conducted experiments on various other examples, including objects without heads, fine-designed text, and multiple objects, as detailed in the figures in the rebuttal file. The baseline method is ProlificDreamer, since it is one of the best text-to-3D methods currently. Our method can achieve spatially consistent results when dealing with objects without heads and fine-designed text. For the Janus Problem of multiple objects, as mentioned in the limitations section of the *Main Paper*, this is constrained by the text-to-3D multi-object generation capability, which will be a direction for our future research.

## Prompt Library

"a bald eagle carved out of wood", "a beagle in a detective's outfit", "a beautiful rainbow fish", "a bichon frise wearing academic regalia", "a cat with a mullet", "a ceramic lion", "a chihuahua wearing a tutu", "a chimpanzee holding a peeled banana", "a chimpanzee looking through a telescope", "a confused beagle sitting at a desk working on homework", "a corgi taking a selfie", "a cute steampunk elephant", "a DSLR photo of a baby dragon drinking boba", "a DSLR photo of a cat wearing a bee costume", "a DSLR photo of a corgi puppy", "a DSLR photo of a dog made out of salad", "a DSLR photo of a frog wearing a sweater", "a DSLR photo of a humanoid robot using a laptop", "a DSLR photo of a lion reading the newspaper", "a DSLR photo of a mouse playing the tuba", "a DSLR photo of a pig playing a drum set", "a DSLR photo of a robot dinosaur", "a fox playing the cello", "a highland cow", "a lionfish", "a pig wearing a backpack", "a red panda", "a tiger playing the violin", "a zoomed out DSLR photo of a baby dragon", "a zoomed out DSLR photo of a monkey riding a bike"

*We sincerely thank all the reviewers once again. For the detailed responses to each reviewer's questions, please refer to the individual replies.*

---

> ### Author Response · Authors · 2024-08-12
> **Happy to have further discussion**
>
> Dear Reviewer sRbF, FWvW, and wiP5,
>
> Thank you for your efforts in reviewing our paper! We greatly appreciate the valuable feedback provided and carefully posted our response. We understand that the reviewers may have very busy schedules. However, we are happy to have further discussion and answer any further questions in the Author-reviewer discussion phase. Looking forward to your feedback!
>
> Best,
>
> Authors of submission 898

---

### Author Response · Authors · 2024-08-13
**Still happy to have further discussion**

Dear Reviewers sRbF, FWvW, and wiP5,

As today marks the final day of the Author-Reviewer Discussion phase, we are still here and welcome any further discussion or feedback you may have on our paper. We sincerely look forward to your feedback. Your insights are incredibly valuable to us.

Thank you sincerely for all the time and effort during the review process.

Best,

Authors of submission 898

---

> ### Author Response · Authors · 2024-08-14
>
> Dear Reviewers,
>
> Thank you for your efforts and valuable suggestions! We will continue revising the paper based on your feedback. Thank you once again to everyone!
>
> Best regards,
>
> Authors of submission 898

---

### Decision · Program_Chairs · 2024-09-25

**Decision:**

Accept (poster)

**Comment:**

This paper addresses the Janus Problem in SDS-based text-to-3D methods by introducing the LCGen, which utilizes varied certainty priors across viewpoints to ensure view-consistent generation. The approach is designed to mitigate view inconsistencies without additional data or computational overhead, effectively integrating into existing text-to-3D frameworks.

The method stands out for its simplicity and effectiveness, as highlighted by reviewers. It innovatively tackles the root causes of the Janus Problem without the need for extra data or significant computational resources, aligning well with the current SDS framework. The empirical results, as noted, validate the approach’s ability to improve view consistency in generated 3D models.

Reviewers consistently pointed out the lack of comprehensive comparisons with existing methods, which could strengthen the paper’s impact. There is also concern about the method's performance in complex scenarios not covered in the study, such as objects without a clear 'head' orientation or those requiring fine-grained details. Enhancing the dataset diversity used in experiments and providing more detailed quantitative analyses could address these concerns. Moreover, expanding the discussion on the method's limitations and potential failure cases would provide a clearer understanding of its applicability and scope.